# Improvement of Interlayer Adhesion and Heat Resistance of Biodegradable Ternary Blend Composite 3D Printing

**DOI:** 10.3390/polym13050740

**Published:** 2021-02-27

**Authors:** Wattanachai Prasong, Akira Ishigami, Supaphorn Thumsorn, Takashi Kurose, Hiroshi Ito

**Affiliations:** 1Graduate School of Organic Materials Science, Yamagata University, 4-3-16 Jonan, Yonezawa, Yamagata 992-8510, Japan; wattanachai.pra@gmail.com (W.P.); akira.ishigami@yz.yamagata-u.ac.jp (A.I.); 2Research Center for GREEN Materials and Advanced Processing (GMAP), 4-3-16 Jonan, Yonezawa, Yamagata 992-8510, Japan; thumsorn@yz.yamagata-u.ac.jp (S.T.); takashi.kurose@yz.yamagata-u.ac.jp (T.K.)

**Keywords:** anisotropic, biodegradable polymers, composites, FDM 3D printing, interlayer adhesion, heat resistance

## Abstract

Poly(lactic acid) (PLA) filaments have been the most used in fused deposition modeling (FDM) 3D printing. The filaments, based on PLA, are continuing to be developed to overcome brittleness, low heat resistance, and obtain superior mechanical performance in 3D printing. From our previous study, the binary blend composites from PLA and poly(butylene adipate-co-terephthalate) (PBAT) with nano talc (PLA/PBAT/nano talc) at 70/30/10 showed an improvement in toughness and printability in FDM 3D printing. Nevertheless, interlayer adhesion, anisotropic characteristics, and heat resistance have been promoted for further application in FDM 3D printing. In this study, binary and ternary blend composites from PLA/PBAT and poly(butylene succinate) (PBS) with nano talc were prepared at a ratio of PLA 70 wt. % and blending with PBAT or PBS at 30 wt. % and nano talc at 10 wt. %. The materials were compounded via a twin-screw extruder and applied to the filament using a capillary rheometer. PLA/PBAT/PBS/nano talc blend composites were printed using FDM 3D printing. Thermal analysis, viscosity, interlayer adhesion, mechanical properties, and dimensional accuracy of binary and ternary blend composite 3D prints were investigated. The incorporation of PBS-enhanced crystallinity of the blend composite 3D prints resulted in an improvement to mechanical properties, heat resistance, and anisotropic characteristics. Flexibility of the blend composites was obtained by presentation of PBAT. It should be noted that the core–shell morphology of the ternary blend influenced the reduction of volume shrinkage, which obtained good surface roughness and dimensional accuracy in the ternary blend composite 3D printing.

## 1. Introduction

In recent years, three-dimensional (3D) printing technology has seen rapid growth. In particular, the fused deposition modeling (FDM) method has been popular in applications because it is inexpensive for thermoplastic materials and low maintenance [1,2,3,4]. Moreover, it has been used in prototypes and the production of custom parts in many industries including the medical, food, textile, automotive, aerospace, and construction industries [5,6,7,8,9,10]. In FDM 3D printing, melted thermoplastic is extruded in the X, Y, and Z directions, and laminated layer by layer on a platform. Products from the laminated layers have anisotropic behavior, in which mechanical properties of the products in a vertical direction (z-direction) are weaker than in the others. Laminated layers are cooled suddenly when building on the platform, resulting in poor interlayer adhesion. These disadvantages of FDM 3D printing have been overcome by controlling materials and processing conditions [11,12,13]. 

Thermoplastic materials, such as poly(lactic acid) (PLA) and acrylonitrile-butadiene-styrene (ABS), are widely used in FDM 3D printing [14,15,16,17]. PLA is most popular in 3D printing because of its biodegradability, high strength, and environmental friendliness. However, PLA has limitations of brittleness and low heat resistance. Research has shown that blending with ductile polymers, i.e., poly(butylene adipate-co-terephthalate) (PBAT), poly(butylene succinate) (PBS), polycaprolactone (PCL) [14,15,16,17,18,19,20,21,22,23,24], and incorporating with particulate fillers and reinforcing fibers [25,26,27,28,29], can improve the limitations of PLA. From our previous study, the binary blend composites from PLA and PBAT blends (PLA/PBAT) and composites with nano talc (PLA/PBAT/nano talc) at 70/30/0/10 exhibited good toughness and printability, and can be presented as alternative filaments in FDM 3D printing [28]. Nevertheless, interlayer adhesion, anisotropy, and heat resistance should be developed for various applications. 

Biodegradable poly(butylene succinate) (PBS) shows excellent processability and biodegradability, and better heat resistance compared to other bio-based polymers such as PLA and PBAT [30,31]. It has been reported that PBS can promote crystallization rate, act as a nucleating agent of PLA, and induce crystallinity in PLA/PBS blends [19,20,21,22,32,33,34,35]. The improvement to polymer crystallinity influenced an enhancing of heat resistance in PLA/PBS blends and PBS/PBAT blends [31,36]. Additionally, Ou-Yang et al. explained that FDM printing with PBS/PLA blends showed good interlayer bonding when increasing PBS content, which was due to a decrease in zero-shear viscosity of the melt [34]. Wu et al. and Ravati et al. showed that they could balance the stiffness and toughness of final products of biodegradable ternary blends from PLA/PBAT/PBS [21,24]. Hence, the development of PLA/PBAT/nano talc with PBS will be an alternative to biodegradable material in FDM 3D printing.

It is interesting to note that the morphology of polymer blends significantly influences the performance of the blend system [20,21,22,23,24,25,36,37,38,39,40,41]. The morphology and phase inversion of immiscible polymer blends can be estimated from the polymer viscosity and volume fraction of the components. The morphology of binary polymer blends generally has a matrix, a dispersed phase morphology, and a co-continuous structure [36,37,39]. Conversely, the morphology of ternary blends is complex, which has been predicted using the spreading coefficient from Harkin’s theory [20,23,24,37,38,39,40,41]. Typical morphologies of ternary blends are classified as (a) two separated phases, (b) core–shell structures, and (c) partial encapsulation, according to the spreading coefficient prediction from the interfacial tension of polymer pairs [37,38]. Core–shell structures in ternary blends have been predicted not only by the spreading coefficient (SC) but also the relative interfacial energy (RIE) and the dynamic interfacial energy (DIE), and have been confirmed by experiments [39]. Ravati et al. reported wetting of ternary blends from PLA, PBAT, PBS, and PCL [23,24]. The interfacial tension of the polymer pair was calculated by the harmonic mean equation. Then, the SC was used to predict wetting and morphologies of ternary blends. Additionally, the viscosity and flow behaviors of molten polymer during extrusion can inform printability and dimensional accuracy in 3D printing [42,43].

This research aimed to prepare binary and ternary biodegradable PLA/PBAT/PBS blends with nano talc for FDM 3D printing. Mechanical properties, heat resistance, interlayer adhesion, and 3D printability (the dimensions and appearances of finished products) of the blend composites were investigated. The influence of PBS on properties and printability of the ternary blend composite was analyzed using the internal structure, viscosity, and blend morphology, and the anisotropy of the 3D print was obtained. 

## 2. Materials and Methods 

### 2.1. Material

Fully biodegradable plastics were used for ternary blends in FDM 3D printing. PLA (Luminy^®^ PLA L175) was supplied from Total Corbion PLA (Rayong, Thailand) Ltd. PBAT (ecoflex^®^ F Blend C1200) was supplied from BASF Japan Ltd. (Tokyo, Japan). PBS (BioPBS™ FZ91PM) was supplied from PTT MCC Biochem Co., Ltd., Bangkok, Thailand. The melt flow rates at 190 °C of PLA, PBAT, and PBS were 3, 2.7–4.9, and 5 g/10 min, respectively. Nano talc powder (nano ACE D-800, particle size 800 nm) was supplied from Nippon Talc Co., Ltd., Osaka, Japan.

### 2.2. Compounding and Preparation of 3D Printing Filaments

Biodegradable binary and ternary blends from PLA/PBAT/PBS were prepared at a ratio of 70:30, where PLA was used as a matrix at 70 wt. % and blended with PBAT and PBS at 30 wt. %. The PLA/PBAT/PBS blends were compounded with nano talc 10 wt. %. The compositions of PLA/PBAT/PBS/nano talc composites are defined in Table 1. PLA, PBAT, and PBS were dried in an oven at 100 °C for at least 6 h. The blend composites were compounded in a twin-screw extruder (KZW15TW-30MG-NH, Technovel Co., Ltd., Tokyo, Japan, L/D = 45) at barrel temperature of 170–200 °C and screw speed of 100 rpm. The blend composites were pelletized and dried at 100 °C for at least 6 h before being applied to the filament (1.75 ± 0.05 mm) using a capillary rheometer (Capilograph 10, Toyo Seiki Seisaku-sho, Ltd., Tokyo, Japan). The conditions were set at a temperature of 180 °C with an extruded speed of 40 mm/min and drawing speed of 1.10 m/min. 

### 2.3. Preparation of 3D-Printing Products

3D-printing products were designed and exported as a standard triangle language (STL) file type in Solid Works 2017 software in a dumbbell shape according to ISO 527-2 type 1BA for tensile testing, and a bar shape according to ISO 178 for flexure testing, as shown in Figure 1. The blend composites were printed using an FDM 3D printer (da Vinci 1.0 Pro, XYZ printing, Inc., New Taipei City, Taiwan) with a nozzle size of 0.4 mm. The printing conditions are shown in Table 2.

### 2.4. Characterization

#### 2.4.1. Thermal Properties and Heat Resistance 

Thermal behavior and crystallization of 3D-printed specimens were evaluated by differential scanning calorimetry (DSC Q200, TA Instruments Inc., New Castle, DE, USA). The sample of 5.0–10.0 mg was prepared from the middle of horizontal dumbbells. The temperature was set at −70–200 °C with a heating rate of 10 °C/min with heat–cooling–heat cycles and held isothermally for 5 min between heating and cooling cycles. The degree of crystallinity (*X*_c_) was calculated using the following equation: (1)$$X_{c}\;\left(\%\right)=\frac{\Delta H_{m}-\Delta H_{cc}}{\Delta H_{f}^{0}}\times \frac{1}{W}\times 100$$
where Δ*H_m_* is the enthalpy of melting, Δ*H_cc_* is the enthalpy of cold crystallization, Δ*H*^0^*_f_* is the heat fusion of fully crystalline PLA 93.7 J/g, and *W* is the weight fraction of PLA in the blend composites [19]. 

Dynamic mechanical analysis (DMA) was investigated using a dynamic mechanical analyzer (RSA-G2, TA Instruments, New Castle, DE, USA). The size of the specimen was 6 mm × 2 mm × 30 mm (width × thickness × length). The specimen was tested in a three-point bending mode at temperature ranges of 30–120 °C at a strain rate of 0.1%, frequency of 1 Hz, and heating rate of 3 °C/min.

#### 2.4.2. Mechanical Properties

For the dumbbell and bar shapes of the 3D-printed specimens, both horizontal and vertical directions, tensile and flexural properties were investigated using a universal testing machine (Strograph VG, Toyo Seiki Seisaku-sho, Ltd., Tokyo, Japan) at a testing speed of 10 mm/min for tensile testing and 2 mm/min for flexural testing according to ISO 527 and ISO 178, respectively.

#### 2.4.3. Morphology

The morphology of cryogenic fractured surfaces of compression-molded PLA/PBAT/PBS blends composites was observed to clarify the structure of the ternary blends by scanning electron microscopy (SEM, JSM-6510, JEOL Ltd., Tokyo, Japan). The PBAT phase in the ternary PLA/PBAT/PBS blends was etched using tetrahydrofuran (THF) to identify the polymer phase in the blends [24]. The samples were coated with platinum before observation.

The morphology of the cryogenic fractured surfaces of the 3D-printing dumbbells was examined by SEM (TM3030plus, Hitachi, Ltd., Tokyo, Japan). SEM images were used to analyze the voids area using Image J software.

#### 2.4.4. Rheological Behavior 

The rheological properties of the blend composites were measured by a rotary rheometer (Modular Compact Rheometer, MCR 302, Anton Paar GmbH, Graz, Austria) with a parallel plate Ø25 mm. Complex viscosity was analyzed with angular frequency (ω) 0.01 to 1000 rad/s at 210 °C, and 1.0% strain.

#### 2.4.5. Surface Roughness

The roughness (R_a_) of horizontal and vertical 3D-printed dumbbell specimens was observed using a 3D optical surface profiler (NewView 8300, Zygo Corporation, Middlefield, CT, USA). The observation was investigated in the middle area of the specimen on the width and the thickness side by 10× on the top field dimension area of 1600 × 1600 μm^2^ with a scan length of 150 μm.

#### 2.4.6. Dimensional Accuracy

Dimensional accuracy of the 3D-printed specimen was observed from the bar shape. The average measurements of thickness, width, and length of the 3D-printed specimens were measured 5 times using a digital micrometer (Mitutoyo, Coolant Proof IP65-MX, range 0–25 mm, and accuracy ± 1 µm) and a digital Vernier caliper (Niigata Seiki, DT-150, range 0–150 mm, accuracy ± 0.01 mm). The measured dimensions were compared with the original dimensions and a deviation was calculated using the following equation [44]:(2)$$Dimensional\;deviation\;\left(\%\right)=\frac{Original\;dimension-Actual\;dimension}{Original\;dimension}\times 100$$
where the original dimension is the dimension from computer-aided design (CAD) (10 mm wide, 2 mm thick, and 60 mm long) and the actual dimension is the dimension from the average measurement. 

## 3. Results and Discussion

### 3.1. Thermal Properties and Heat Resistance of PLA/PBAT/PBS/nano talc Composites 3D Prints

#### 3.1.1. Thermal Properties and Crystallization Behavior 

Figure 2 shows DSC thermograms of horizontal PLA/PBAT/PBS/nano talc composites 3D prints from the first heating cycle in Figure 2a, the cooling cycle in Figure 2b, and the second heating cycle in Figure 2c. The thermal properties of neat polymers and the blend composite 3D prints are summarized in Table 3. At first heating, glass transition temperatures (T_g_) of PLA matrix in the blend composites were about 59–60 °C, and unchanged as compared to neat PLA. Melting peaks were at about 112 °C and 174 °C, which represented the melting temperatures (T_m_) of PBS and PLA, respectively. Nevertheless, the melting peak of PBAT was influenced by cold crystallization and was unnoticeable from the blend composites. The melting peaks were related to neat polymers, and did not shift to each other, which informed the immiscibility of these PLA/PBAT/PBS blend composites [19,20,21]. Cold crystallization temperature (T_cc_) was found only in the first heating scan, which informed incomplete crystallization during 3D printing. However, the cold crystallization temperature and its enthalpy (Δ*H_cc_*) of the composites decreased when adding PBS, as shown in Figure 2a and Table 3. Correspondingly, PBS acted as a nucleating site and induced the crystallization of PLA in blend composites. PBS would influence on organized polymer crystallization during the cooling or solidification of the 3D printing process and increased the degree of crystallinity (*X_c_*) of PLA [19,34] as presented in Table 3. The crystallization temperature (T_c_) of PLA in the blend composites was shown sharply at about 113 °C to 117 °C, as shown in Figure 2b, which confirmed an improvement of PLA crystallization and its effects on the stabilized printability of 3D-printed products. As shown in Figure 2c, the cold crystallization disappeared after the second heating and revealed double melting peaks of the PLA matrix in the blend composites. It was considered that the addition of PBAT, PBS, and nano talc enhanced PLA crystallization and affected the recrystallization and remelting of PLA crystal in the composites [22,28]. Therefore, the incorporation of PBS enhanced the degree of crystallinity of 70/0/30/10 and 70/10/20/10 composite 3D prints, which should improve the heat resistance and mechanical properties of the binary and ternary blend composites.

#### 3.1.2. Dynamic Mechanical Properties and Heat Resistance

The dynamic mechanical properties and heat resistance of binary and ternary blend composite 3D prints were analyzed using dynamic mechanical analysis. Figure 3 shows the dynamic mechanical properties, storage modulus, and Tan δ of the blend composites compared with neat PLA, and the results are tabulated in Table 4. The storage modulus informs the elasticity and stiffness of the materials. The onset temperature from storage modulus and the temperature from Tan δ peak indicated a glass transition temperature of the polymer. The storage modulus of the neat PLA and the blend composite 3D prints presented a glassy stage at below the glass transition of PLA and a sudden drop to a rubbery state when temperature increased over T_g_ [17], as presented in Figure 3a. The incorporation of PBS increased the storage modulus of the composite 3D prints, which exhibited higher storage modulus at all temperature ranges compared to the binary blend with PBAT (70/30/0/10) and neat PLA, as presented in Figure 3a. 

It can be indicated that PBS improved heat resistance of the blend composite 3D prints, especially when compared to neat PLA. The storage moduli of the PLA/PBAT/PBS/nano talc composites were 3.0 GPa and 2.6 GPa for 70/0/30/10 and 70/10/20/10, respectively, which were higher than neat polymer, as summarized in Table 4. It was considered that the addition of PBS and nano talc improved elasticity and stiffness of the blend composite 3D prints, which was due to an increase of these blend composites’ crystallinity. The onset temperature from the storage modulus curve decreased the blend composite 3D printing. However, the glass transition temperatures (from Tan δ) of the blend composite 3D printing unchanged. In addition, Tan δ peak intensities decreased with increasing PBS contents, as shown in Figure 3b, which informed the increment of stiffness and the reduction to damping properties in the blend composite [17].

### 3.2. Mechanical Properties of PLA/PBAT/PBS/nano talc Composites 3D Prints

Tensile and flexural properties of the PLA/PBAT/PBS/nano talc composites 3D prints in the horizontal and vertical directions are presented in Table 5 and Table 6, respectively. From Table 5, the tensile strength and Young’s modulus of the horizontal and the vertical blend composite 3D prints significantly increased when adding PBS. This was due to compatibility between polymer blends and PBS. In addition, nano talc promoted a degree of crystallinity of the composites, which improved the strength and stiffness of the blend composite 3D printing [18,19]. The elongation at break of the horizontal PLA/PBAT binary blend composite (70/30/0/10) showed highest elongation at break. However, the ductility of the PLA/PBAT/PBS ternary blend was maintained, which was considered to be due to the addition of high-flexibility PBAT [18]. Table 6 tabulates the flexural properties of the blend composite 3D prints, which were improved by the addition of PBS and the presence of PBAT and nano talc. It can be noted that the addition of PBAT yielded the ductility of the blend composites, which obtained higher elongation at break than the binary blend composites of the 70/0/30/10. The incorporation of PBS improved tensile strength, Young’s modulus, flexural strength, and flexural modulus of the composite 3D prints. Nevertheless, the tensile and flexural properties were poor in the vertical direction because of the anisotropic characteristics in FDM 3D printing, where 3D printing products had low resistance to tensile and bending loads in the vertical printing direction (z-direction) [11,28,45].

The strength acquired by the vertical specimen can be considered to be due to the adhesion strength between layers [1,34]. From the results of the vertical 3D prints, the incorporation of PBS increased the tensile and flexural strength of the vertical 3D prints as presented in Table 5 and Table 6. This should inform an improvement to the adhesion strength and the anisotropy of these blend composite 3D prints. Therefore, the anisotropic ratio on mechanical properties can be calculated by using the following equation [46]:(3)$$Anisotropic\;ratio\;=\frac{\left|S\;\left(horizontal\right)-S\;\left(vertical\right)\right|}{S\;\left(horizontal\right)}$$
where *S* is the tensile strength and flexural strength of the horizontal and vertical specimen.

Figure 4 depicts the anisotropic ratio of the PLA/PBAT/PBS/nano talc composite 3D prints. The anisotropic ratio in tensile and flexural strengths decreased when incorporated with PBS in the 70/0/30/10 and 70/10/20/10 as compared to 70/30/0/10. This result influenced the retention of the anisotropic characteristic of the tensile and flexural strength. It is worth noting that the anisotropic ratio of the 70/10/20/10 3D printing was lowest. It can be noted that the ternary blend composite 70/10/20/10 shows a reduction in the anisotropy with its balance in strength and stiffness from adding PBS and nano talc, and ductility from PBAT content. This might be due to good interlayer adhesion and surface integrity of the 3D print, which could promote the mechanical properties of the 70/10/20/10 3D prints [47,48,49]. 

### 3.3. Complex Viscosity of PLA/PBAT/PBS/nano talc Composite 3D Prints

The rheological behavior was analyzed to explain the morphology of polymer blend composites and flowability in the 3D-printing process. Figure 5 and Appendix A present the complex viscosity at 210 °C of neat polymers, and the blend composites as functions of angular frequency and shear rate, respectively. The complex viscosity (η^∗^) of neat polymers and the blend composites decreased with increasing angular frequencies as well as shear rates. However, the viscosity of neat PLA was almost as constant as Newtonian behavior, while the others revealed non-Newtonian behavior. The power-law model has been used to classify non-Newtonian fluid behavior, as shown in Equation (S1) [27,36,42]. The power-law index (n) can be influenced by a Newtonian fluid when n = 1, by a shear thinning fluid when n < 1, and by a shear thickening fluid when n > 1 [42]. Appendix A tabulates the determination of flow behavior of neat polymers and the blend composites. From the power-law index, PLA is closed to the Newtonian behavior (n = 0.98), whereas the others are shear thinning behavior (n < 1). From Figure 5a, the complex viscosity of the blend composites was larger than neat PLA and PBAT, which was due to the restriction of molecular movement by the addition of nano talc and the long chain of PBAT and PBS. From the results, the complex viscosity of neat PBS was higher than neat PLA and PBAT, which influenced the increment of the viscosity with the addition of PBS, especially when nano talc existed in the 70/0/30/10 and 70/10/20/10 blend composites [21,28,50] as shown in Figure 5b. It can be informed that the viscosity at high frequency and shear thinning behavior can be related to the flowability and printability in the 3D-printing process [42,43]. Rastin et al. reported the benefit of viscosity and shear thinning behavior in 3D printing [42]. From this research, the material showed shear thinning behavior, but has low viscosity resulting in a lack of printability. On the contrary, the material with a higher viscosity exhibiting shear thinning behavior could obtain higher shape accuracy after printing. The viscosity of the ternary blend composite was lower than the binary blend composite and may benefit for promoting the adhesion between layers in the 3D-printing process [34]. Additionally, in the PLA/PBAT/PBS blend system, Jazani et al. showed that a lower viscosity phase as a core structure can be encapsulated with a higher viscosity phase as a shell to perform a core–shell structure on the polymer matrix, which may facilitate polymer chains in the ternary blend composite [40]. 

### 3.4. Interlayer Adhesion and Morphology of PLA/PBAT/PBS/nano talc Composite 3D Prints

#### 3.4.1. Interlayer Adhesion of PLA/PBAT/PBS/nano talc Composite 3D Prints

Interlayer adhesion is one of the most important properties of FDM 3D printing, where the development of the interlayer adhesion may improve mechanical properties [13,14]. In this study, interlayer adhesion was observed from the cross-sectional surfaces of the blend composite 3D prints, as shown in Figure 6a–c for horizontal printing and Figure 6d–f for vertical printing. From Figure 6, layer-by-layer deposition can be seen from 3D printing with oval shapes of about 0.2 mm of the layer thickness and 0.4 mm of the layer width. The 3D printing layers seem to adhere well to the thickness in all binary and ternary blend composites from both the horizontal and the vertical 3D prints. However, voids exist between the layers, which was due to polymer melt rapidly solidifying when deposited in the 3D-printing process. SEM was used to analyze the void area using the Image J software, and presented in Table 7. At higher magnification of horizontal 3D printing in Figure 6a–c, the air void in the middle area of the ternary blend composite 70/10/20/10 was smaller than the other, with the lowest void area of 0.05 mm^2^ as shown in Table 7. It was considered that the incorporation of PBS improved the crystallinity of the blend composite, which influenced layer shrinkage when the printed resulting in the reduction of the air void between the layers [19,34]. Conversely, the highest void area was the binary blend composite of 70/0/30/10, which was considered to be due to the reduction of the filament welding [18]. In addition, the relationship between viscosity, crystallization temperature, and degree of crystallinity contributed to the promotion of the interlayer adhesion of the 3D printing process [50].

From Figure 6d–f, it can be observed that the interlayer adhesion of the vertical dumbbell was improved, with the void area clearly reduced. It was thought that the printing area in the middle of the vertical dumbbell had a short cycle time for the layers solidified and adhered with the high temperature of the previous layers [39]. Therefore, following the high temperature of the previous layer, the bonding between the layers was developed from molecular entanglement [34,47,51]. By observing the high magnification of vertical specimens, the void areas were smaller than the horizontal direction as presented in Table 7, and the triangle voids became round, as observed from Figure 6. It can be noted that the addition of PBS increased the crystallinity of the blend composites, which influenced the layer shrinkage and resulted in enhanced interlayer adhesion. However, the incorporation of higher PBS content would raise more layer shrinkage that causes warpage, poor surface finish, and lower dimensional accuracy of the 3D-printed products [1,19,34]. 

#### 3.4.2. Morphology of PLA/PBAT/PBS/nano talc Composites

In this research, we developed 3D printing filaments from the binary and ternary blends composited with nano talc. The compatibility and morphology between the polymer blends and nano talc composites influenced the mechanical properties and the interlayer adhesion of the composite 3D prints. We observed the morphology of the blend composites from the cryogenic fractured surface of the compression mold specimen. 

Figure 7 shows SEM images of the blend composites at a magnification of ×8000. These images demonstrate that the nano talc particle, as indicated by arrows, was well distributed on the polymer matrix. The nano talc as a platelike additive has been used to improve stiffness, enhancing thermal resistance and increasing the nucleating ability of the polymer composites [27]. The PLA/PBAT/nano talc composite (70/30/10) 3D-print has a superior tensile modulus and improved printability [28]. From Figure 7, the nano talc seemed to be well distributed on the polymer blends, which significantly improved the crystallization of these polymer blend composites due to heterogeneous nucleating sites from the nano talc particles. Hence, the nano talc enhanced the crystallinity, the interlayer adhesion, and the mechanical properties of the blend composite 3D prints. However, the affinity of the nano talc to the PLA, PBAT, or PBS phase was not clear, and should be studied further.

From Figure 7, it was difficult to observe the distribution of the polymer minor phases on the PLA matrix of the ternary blends from PLA, PBAT, and PBS with nano talc composite. Consequently, we observed the morphology of binary and ternary PLA/PBAT/PBS blends without nano talc. The morphology before etching PBAT is presented in Figure 8 (top row) and after etching PBAT in Figure 8 (bottom row). SEM images of PLA70/PBAT30, PLA70/PBS30, and PLA70/PBAT10/PBS20 blends are shown in Figure 8a–c, respectively. PLA as the major content was the matrix of the blends. PBAT and PBS as the minor were the dispersed phase on the PLA matrix, as presented in Figure 8a,b. PBAT and PBS dispersed phase sizes, about 1–3 μm, were uneven, which indicated an immiscibility of polymer blends. On the other hand, PBS and PBAT revealed the core–shell morphology on the PLA matrix and their core–shell dispersed structure looked smaller than the phase size in the binary blends. 

The morphology of the polymer blends at various compositions can be estimated from the phase inversion from Equation (S2), as presented in [36,39]. Appendix A shows the prediction of phase inversion in the binary blends from PLA, PBAT, and PBS. From these materials, PLA is the matrix with dispersed phase of PBAT or PBS. Harkin’s spreading theory has explained this by the effect of interfacial tension between polymer phases on the wetting and the morphology formation in ternary blends [20,23,24,37,38,39,40,41]. Harkin’s spreading equations are presented in Equations (S3)–(S5). For PLA/PBAT/PBS ternary blends, PLA is the continuous phase or the matrix as phase A. PBAT and PBS are dispersed phases as phase B and phase C, respectively. The SC can be calculated from Equations (S3)–(S5). When the coefficient $\lambda_{ABC}\;$is positive, the phase B will encapsulate the phase C [37,38,41]. The interfacial tension between polymer phases can be calculated using the harmonic mean equation as depicted in Equation (S6) [23]. The interfacial tensions of the polymer blends are summarized in Appendix A. In this study, the surface tensions of PLA, PBAT, and PBS are drawn from Ravati and Favis [23]. Appendix A tabulates the SC of the ternary blends PLA/PBAT/PBS. After etching PBAT with THF, it was found that the core phase remained, whereas the shell phase disappeared, as presented in Figure 8c. This could indicate that PBS as the core was encapsulated by the shell of PBAT [23,24]. Jazani et al. and Hemmati et al. [39,40] explained that the phase at low viscosity should encapsulate the phase with high viscosity. Therefore, as tabulated in Appendix A, high viscosity of PBS was encapsulated by low viscosity of PBAT. From Appendix A, the coefficient of $\lambda_{PLA/PBAT/PBS}$ is positive and confirms that PBAT encapsulates PBS in the ternary blends and performs as the core–shell dispersed phase structure [23,24,37,38]. The core–shell of the PBS/PBAT dispersed phase seemed to be smaller than the dispersed PBAT or PBS sizes in the binary blends, which may inform compatibility between PLA, PBS, and PBAT in the blend [20]. In addition, it can be shown that the complete wetting morphology and the reduction of the core–shell diameter in the ternary blends would support polymer printability and showed a positive effect to improve the mechanical properties of 3D-printing composites [40,41] and the balancing of mechanical properties by the incorporation of PBS in ternary blend composites [24].

### 3.5. Surface Roughness and Dimensional Accuracy of PLA/PBAT/PBS/nano talc Composite 3D Prints

#### 3.5.1. Surface Roughness

Surface quality is required in the FDM 3D printing process since surface roughness generally occurs from a filament being deposited layer by layer on the 3D-printed products [1,44,52]. The surface quality of the 3D-printed product can be informed as the surface roughness average (R_a_) on the direction perpendicular to the layer orientation [52]. The surface roughness of the PLA/PBAT/PBS blend composite was observed in the middle of the horizontal and vertical dumbbells on width and thickness sides as depicted in Figure 9a. Figure 9b,c show examples of 3D surface roughness profiles measured from the width side and the thickness side, respectively. The R_a_ of the blend composite 3D prints is summarized in Table 8.

The R_a_ values of the horizontal dumbbells were lower than the vertical dumbbells, especially at the thickness side. It was considered that the solidification of the printed layers in the horizontal printing was more stable than the vertical direction [28,44]. From our previous study, the incorporation of nano talc significantly improved the surface roughness of the vertical PLA/PBAT/nano talc composite 3D prints. However, nano talc influenced fast layer solidification, resulting in an increased void area and decreased interlayer adhesion [28]. In this study, as aforementioned, the void area of the blend composite 3D print decreased with the addition of PBS, resulting in an improvement of interlayer adhesion in the 70/0/30/10 and 70/10/20/10 3D prints as well as decreased of the surface roughness in horizontal 3D printing. This was attributed to the contraction of shrinkage layers by higher crystallinity in these composites with the incorporation of PBS. Nevertheless, at the thickness side, the 70/0/30/10 binary blend composite revealed the highest surface roughness. This might correspond to the limitation of printing resolution at small dimensions [53,54,55]. Hence, the thickness side of a vertical specimen became rounded and deviated from the CAD design (G-code) as presented in Figure 10. Figure 10a draws a schematic of cross-sectional printing lines from the vertical specimen. Yellow lines indicate the number of printed layers, blue lines indicate printed resolution, and red lines are G-code. Figure 10b shows the SEM images of the vertical 70/0/30/10. The curvature and rough surface at the thickness of the 70/0/30/10 vertical 3D printing can be observed. This was due to the expansion of the edge layer (L-3) and incompletely solidified of the previous layer at the narrow area [1,19,47]. Therefore, the edge of the specimen obtained from the vertical deposition was distorted and obtained high surface roughness. It can be noted that the R_a_ of the 70/10/20/10 3D printing was improved, notably the horizontal specimens as presented in Table 8. The result is attributed to the lower layer shrinkage in the 70/10/20/10 compared to the 70/0/30/10, which is due to the core–shell dispersed phase morphology and the decreasing of the air void resulting in the reduction of surface roughness [44,52,56].

#### 3.5.2. Dimensional Accuracy

The final dimensions of an FDM 3D print normally deviates from the CAD dimension designed because of a shrinkage of semi-crystalline thermoplastic filaments. The dimensional deviation in FDM 3D printing is basically from 0.1 to 1.4% and corresponds to the volumetric shrinkage of the thermoplastic [57,58]. The dimensional deviation of blend composite 3D prints was calculated by comparing to the original CAD design according to Equation (2) [44]. Table 9 shows the dimensions and dimensional deviations of the blend composite 3D prints. Figure 11a–c displays the dimensional deviation of width, length, and thickness, respectively. From the results, a positive deviation means dimensions smaller than the original dimensions from the CAD and a negative deviation means dimensions larger than the original dimensions.

The dimension deviation of width and length in the 3D prints was about 0.20% to 0.75% and 0.06% to 0.84%, respectively as displayed in Table 9, and Figure 11a,b. The width and the length of the composite 3D prints with PBS were slightly lower than the original size, which was due to volume shrinkage after complete solidification. The incorporation of talc decreased the coefficient of volume expansion in PLA/PBAT/nano talc composite 3D printing, which improved the volume shrinkage as well as the surface roughness of 70/30/0/10 [28]. However, the PBS enhanced the crystallinity of the composite 3D prints. Thus, crystalline structure orientation during solidification would yield high shrinkage that significantly influences on the dimensional deviation of the blend composite 3D prints with PBS in 70/0/30/10 and 70/10/20/10. It can be noted that the width and the length deviation of both horizontal and vertical 3D prints were less than 1% from the original dimension.

Conversely, Figure 11c shows the thickness deviation of the blend composite 3D prints. The thickness deviation exhibited mainly negative values, meaning that the thickness of the 3D prints was larger than the original dimension. The thickness deviations were about 0.56% to 6.45% higher in the vertical direction than the horizontal one. The maximum thickness deviation was the 70/0/30/10 vertical 3D printing. This corresponded to a large thermal expansion of molten filament along the vertical printing direction where the melted polymer was incompletely solidified from the previous layer at the narrow printing design. From Figure 10a, the shell thickness was set at 2 layers with a 0.2 mm-layer thickness in this work. The inside shell (L-1 and L-2) was first printed, followed by the outside shell (L-3 and L-4), and finally the infill (L-5). L-3 and L-4 were expanded outside following deposition, which was due to a limit from the previous layer (L-1 and L-2) and became an extended edge of the thickness [39]. Therefore, the thickness of the vertical 3D printing was larger than the original dimension and increased the surface roughness of the vertical 3D prints [47,54,56]. This can be indicated that the surface roughness of the 3D prints affected the dimensional accuracy. From Appendix A, the 70/0/30/10 showed the highest viscosity and revealed higher shear thinning behavior as compared to other blend composites. Hence, the printability of the 70/0/30/10 was low, resulting in poor dimensional stability [43].

In addition, the deflection of the 3D printing product is depicted in Figure 12a,b. The highest deflection and surface roughness can be observed in the binary blend composite 70/0/30/10. It was attributed to the crystallinity of the PBS influencing the thermal stress from volumetric shrinkage during a cooling layer onto the lower temperature of the bedplate [19,34]. For this reason, there is poor adhesion between the bottom layer and the bedplate, resulting in warping and deflection in the first layer range of 3D-printed products [59]. From the results, it can be shown that the large volume shrinkage in the binary PLA/PBS blend composite resulted in poor dimensional accuracy and increased distortion of products during printing in the horizontal direction of the 70/0/30/10, as shown in Figure 12a. The thermal expansion influenced the expansion of the edge in the vertical of 70/0/30/10 as presented in Figure 12b. On the other hand, it was considered that the core–shell structure of PBS/PBAT on the PLA matrix influenced the reduction of the polymer layer shrinkage [43]. In addition, the ternary blend showed balancing in viscosity and shear thinning behavior [42,43]. Hence, the ternary blend composite 70/10/20/10 obtained good dimensional accuracy as compared with the binary blend composite 70/30/0/10. It is interesting to note that all binary and ternary blend composite 3D filaments can be printed without warping, as shown in Figure 12c.

## 4. Conclusions

The binary and ternary blend composite filament from PLA/PBAT/PBS/nano talc FDM 3D printing was successfully prepared. The addition of PBS in the blend composite improved polymer crystallinity that enhanced heat resistance and tensile and flexural properties of the 3D prints. The high degree of crystallinity influenced the shrinkage of molten polymer during deposition layer by layer, which reduced void area and increased interlayer adhesion, resulting in the retention of anisotropic characteristics in the blend composite 3D prints with PBS. However, the large volume shrinkage when adding PBS in the binary blend composite exhibited high surface roughness, poor dimensional accuracy, and warping in the 70/0/30/10. Conversely, the ternary blend composites of the 70/10/20/10 revealed a balance of heat resistance and mechanical properties. The addition of PBS and nano talc increased the PLA crystallinity, which improved storage modulus, tensile and flexural strength, as well as anisotropic characteristics. In addition, blending with PBAT obtained ductility in the ternary blend composite. It should be noted that the core–shell of PBS/PBAT on the PLA matrix influenced the reduction of polymer layer shrinkage. The printability and dimensional accuracy were developed by balancing the viscosity and the shear thinning behavior in the ternary blend composite. Therefore, the ternary blend composites of 70/10/20/10 exhibited good surface roughness and dimensional accuracy in FDM 3D printing.

## Figures and Tables

**Figure 1 polymers-13-00740-f001:**
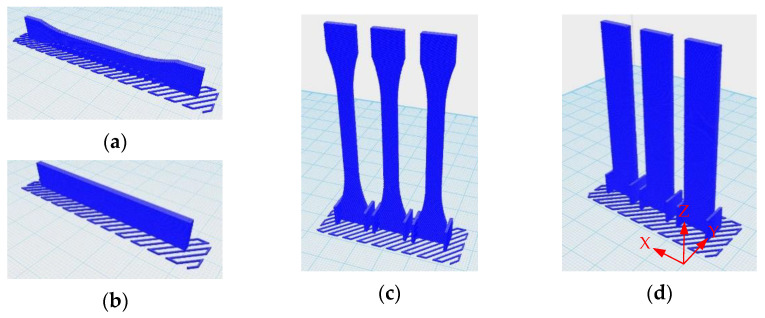
3D printing specimens: (**a**) horizontal dumbbell; (**b**) horizontal bar; (**c**) vertical dumbbell; (**d**) vertical bar.

**Figure 2 polymers-13-00740-f002:**
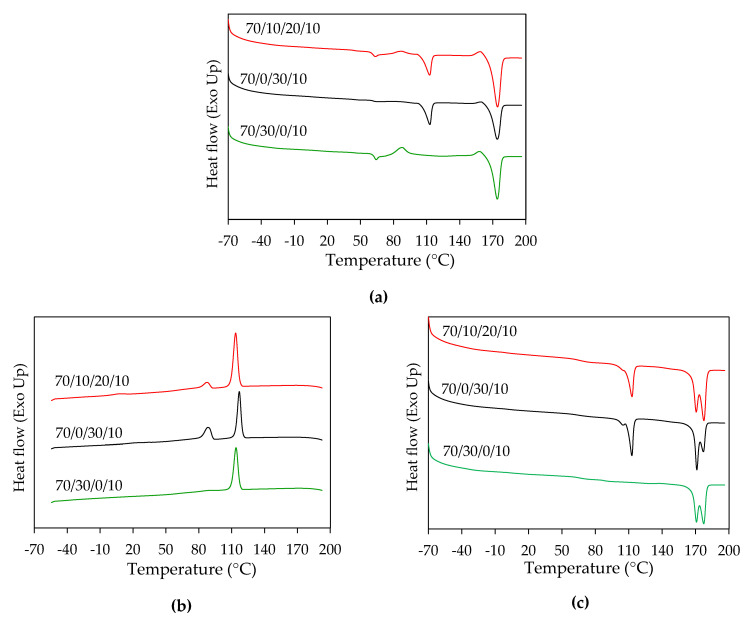
DSC thermograms of PLA/PBAT/PBS nano talc composite 3D prints: (**a**) First heating cycle; (**b**) Cooling cycle; (**c**) Second heating cycle.

**Figure 3 polymers-13-00740-f003:**
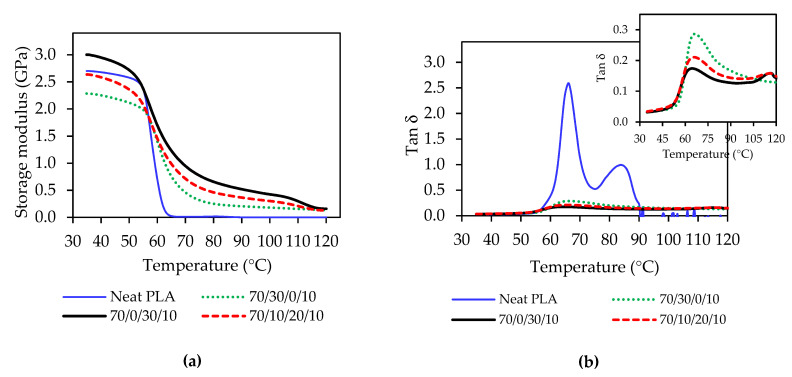
Dynamic mechanical properties of PLA/PBAT/PBS/nano talc composite 3D prints: (**a**) Storage modulus; (**b**) Tan δ.

**Figure 4 polymers-13-00740-f004:**
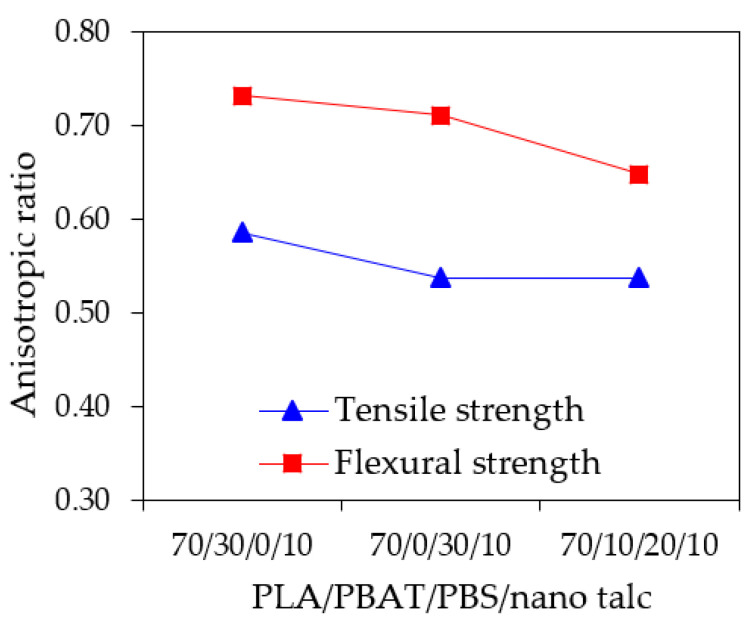
Anisotropic ratio in tensile and flexural strength of PLA/PBAT/PBS/nano talc composite 3D prints.

**Figure 5 polymers-13-00740-f005:**
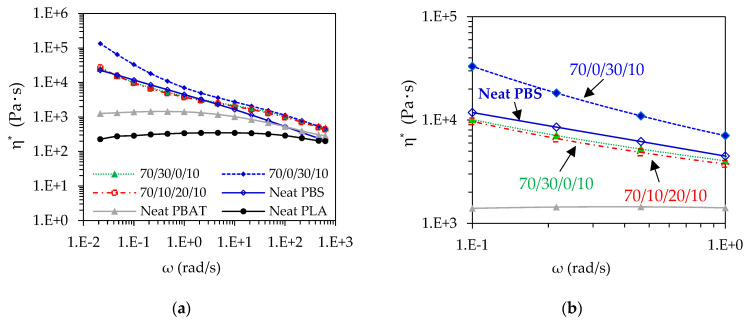
Complex viscosity (η^∗^) of neat polymers and PLA/PBAT/PBS/nano talc composites measured at 210 °C: (**a**) Full image; (**b**) Enlarged image of (**a**).

**Figure 6 polymers-13-00740-f006:**
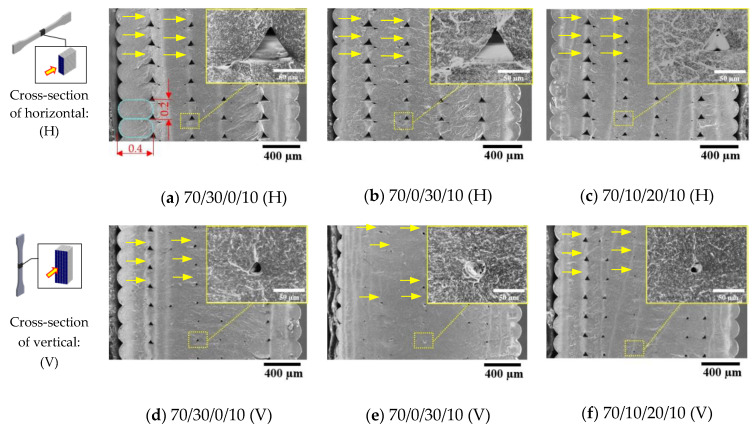
SEM images of the cross-sectional surface of PLA/PBAT/PBS/nano talc composite 3D prints: (**a**–**c**), SEM images from a horizontal dumbbell; (**d**–**f**), SEM images from a vertical dumbbell; (examples of void areas were identified by yellow arrows).

**Figure 7 polymers-13-00740-f007:**
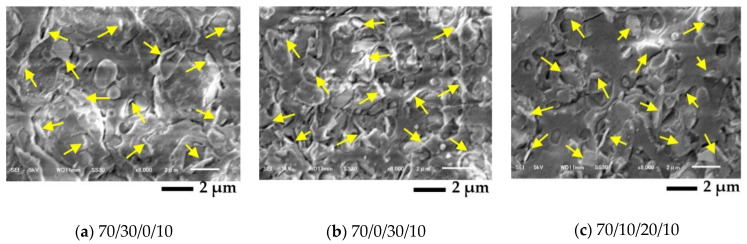
SEM images from cryogenic fractured surface of PLA/PBAT/PBS/nano talc: (**a**) 70/30/0/10; (**b**) 70/0/30/10; (**c**) 70/10/20/10.

**Figure 8 polymers-13-00740-f008:**
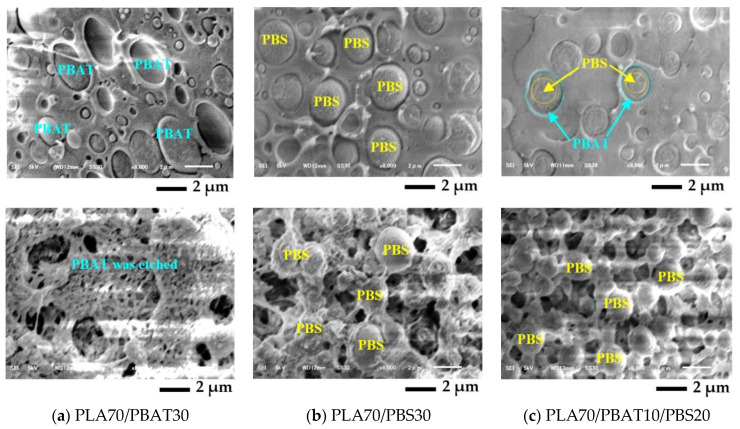
SEM images from the fractured surface of compression-molding binary and ternary blends PLA/PBAT/PBS without nano talc: (**a**) PLA70/PBAT30; (**b**) PLA70/PBS30; (**c**) PLA70/PBAT10/PBS20 (the SEM images in the top row and the bottom row are the fractured surface before and after etching PBAT, respectively).

**Figure 9 polymers-13-00740-f009:**
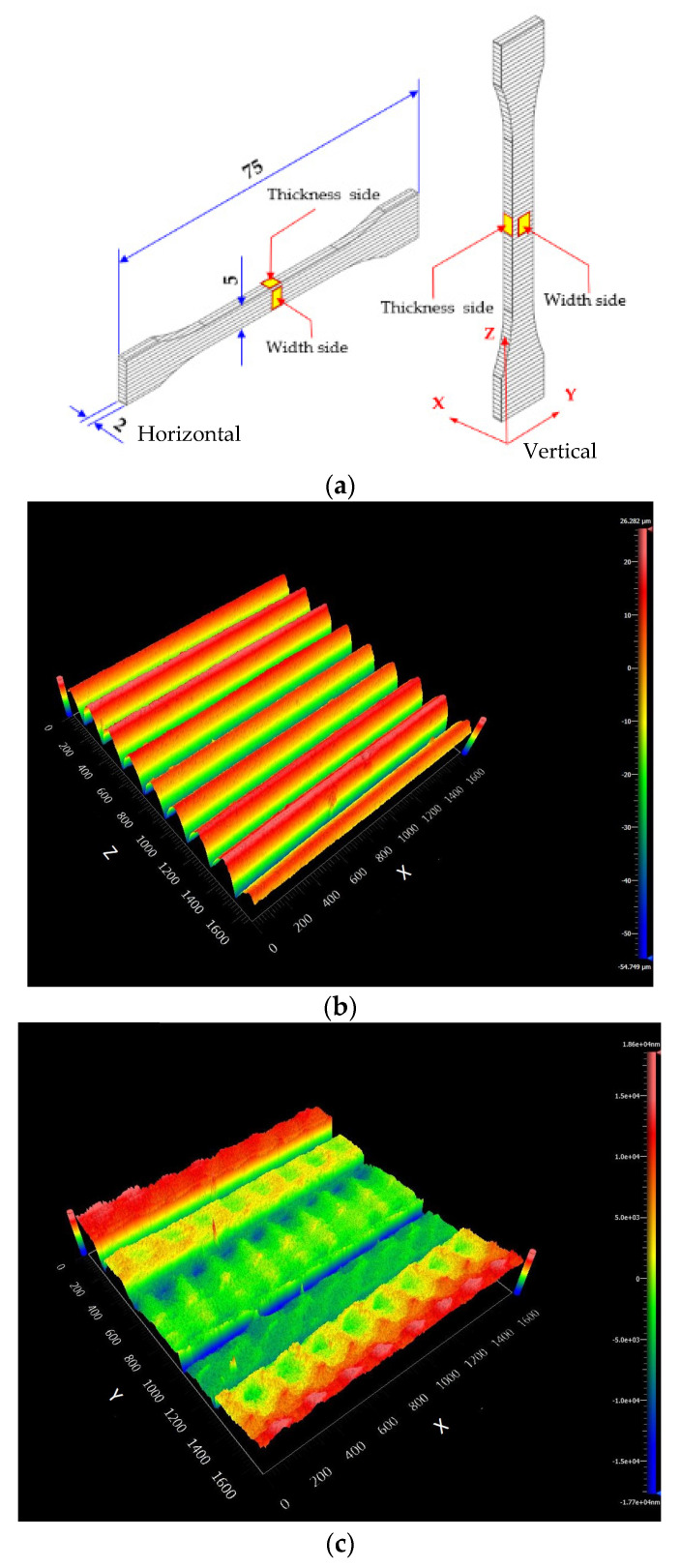
(**a**) Sample observation direction of surface roughness and examples of 3D surface roughness profiles: (**b**) 3D surface roughness profile from width side; (**c**) 3D surface roughness profile from thickness side.

**Figure 10 polymers-13-00740-f010:**
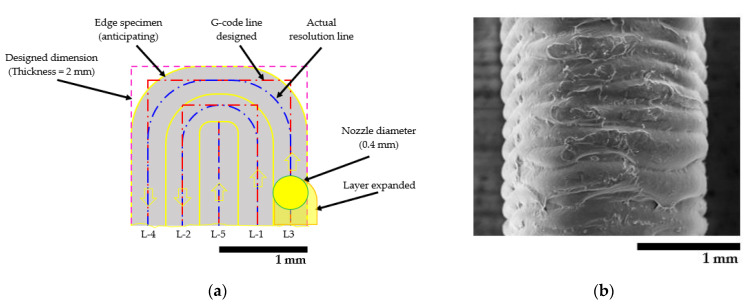
(**a**) Schematic of the vertical 3D printing line; (**b**) SEM image from the thickness side of 70/0/30/10 3D printing.

**Figure 11 polymers-13-00740-f011:**
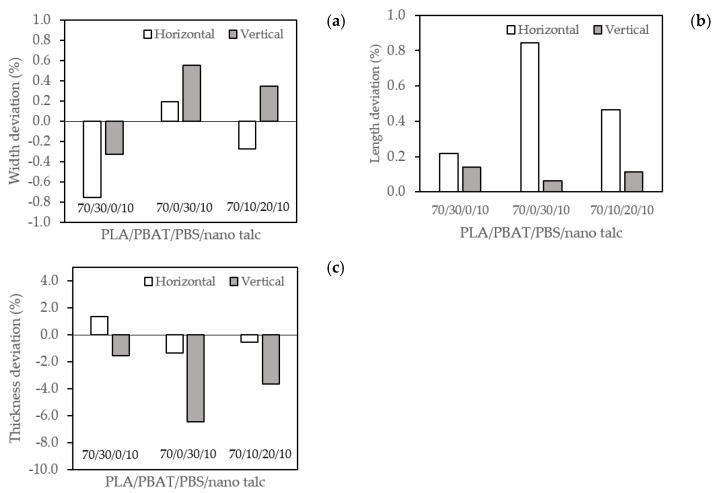
Dimensional deviation of PLA/PBAT/PBS/nano talc composites 3D prints: (**a**) Width deviation; (**b**) Length deviation; (**c**) Thickness deviation.

**Figure 12 polymers-13-00740-f012:**
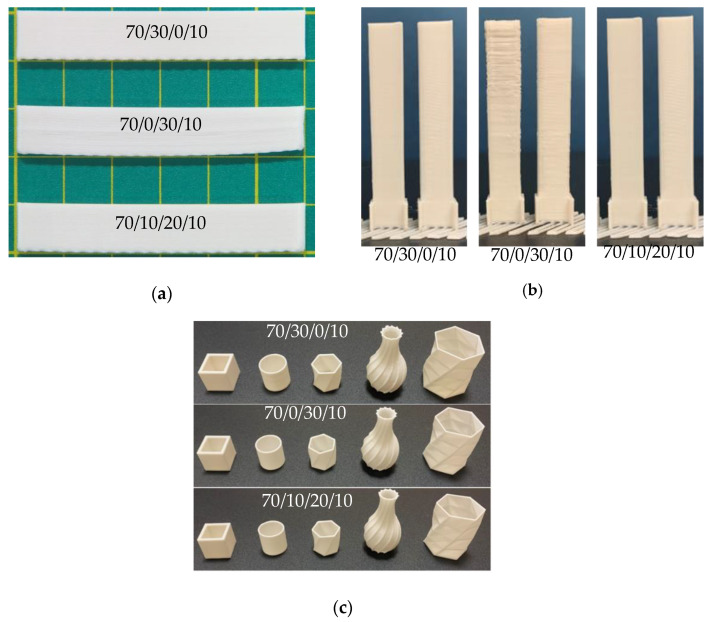
The distortion of 3D printing bars: (**a**) Horizontal direction; (**b**) Vertical direction; and (**c**) 3D printing models of PLA/PBAT/PBS/nano talc composites.

**Table 1 polymers-13-00740-t001:** Compositions of binary and ternary blend composites from PLA/PBAT/PBS/nano talc.

PLA/PBAT/PBS/nano talc	PLA (wt. %)	PBAT (wt. %)	PBS (wt. %)	nano talc (wt. %)
70/30/0/10	63	27	0	10
70/0/30/10	63	0	27	10
70/10/20/10	63	9	18	10

**Table 2 polymers-13-00740-t002:** Conditions of FDM 3D printing.

Parameters	Conditions
Nozzle Temperature	210 °C
Bed Temperature	60 °C
Printing Speed	25 mm/s
Infill Density	100%
Infill Type	Rectilinear
Layer Height	0.2 mm
Shell Thickness	2 layers

**Table 3 polymers-13-00740-t003:** Thermal properties and crystallinity of neat polymers and PLA/PBAT/PBS/nano talc composite 3D prints.

PLA/PBAT/PBS/nano talc	T_c_ (°C)	T_g_ (°C)	T_cc_ (°C)	Δ*H_cc_* (J/g)	PBS T_m_ (°C)	PLA T_m1_ (°C)	PLA T_m2_ (°C)	Δ*H_m_* (J/g)	*X_c_* (%)
Neat materials ^1^									
Neat PLA	-	59.1	-	-	-	174.5	-	7.7	8.24
Neat PBAT ^2^	39.6	−31.0	-	-	-	-	-	20.7	18.2
Neat PBS	86.3	−35.7	-	-	114.2	-	-	69.0	62.6
3D printed dumbbell 1st heating									
70/30/0/10	114.0	60.7	87.3	12.2	-	174.1	-	34.5	37.7
70/0/30/10	116.9	60.1	84.8	2.8	112.9	174.0	-	33.9	52.8
70/10/20/10	113.5	59.4	87.1	5.9	112.7	174.3	-	33.9	47.3
3D printed dumbbell 2nd heating									
70/30/0/10	-	60.4	-	-	-	170.8	177.3	29.8	50.5
70/0/30/10	-	58.3	-	-	112.8	171.1	177.0	30.4	51.5
70/10/20/10	-	60.0	-	-	112.9	170.6	177.3	30.2	51.1

^1^ Neat material data measured from the cooling cycle and the 2nd heating cycle; ^2^ Melting temperature (T_m_) of PBAT is 120.3 °C.

**Table 4 polymers-13-00740-t004:** Dynamic mechanical properties of neat polymers and PLA/PBAT/PBS/nano talc composite 3D prints.

PLA/PBAT/PBS/nano talc	Storage Modulus ^1^ (GPa)	Onset Temperature (°C)	Tan δ Peak (T_g_ PLA) (°C)
Neat PLA ^2^	2.7	58.7	65.9
Neat PBAT ^2^	0.1	-	-
Neat PBS ^2^	0.6	-	-
70/30/0/10	2.4	57.3	66.4
70/0/30/10	3.0	53.8	64.8
70/10/20/10	2.6	54.7	65.6

^1^ Storage modulus was obtained at 35 °C; ^2^ Neat materials were performed by compression molding.

**Table 5 polymers-13-00740-t005:** Tensile properties of PLA/PBAT/PBS/nano talc composite 3D prints.

PLA/PBAT/PBS/nano talc	Tensile Strength (MPa)		Young’s Modulus (GPa)		Elongation at Break (%)	
	Horizontal	Vertical	Horizontal	Vertical	Horizontal	Vertical
70/30/0/10	42.5 ± 1.09	17.6 ± 0.88	0.9 ± 0.02	0.6 ± 0.07	128.2 ± 73.56	3.0 ± 0.22
70/0/30/10	56.9 ± 1.36	26.3 ± 2.55	1.1 ± 0.04	0.8 ± 0.03	6.3 ± 0.47	3.3 ± 0.44
70/10/20/10	50.4 ± 0.97	23.3 ± 0.86	1.0 ± 0.01	0.7 ± 0.03	55.9 ± 19.87	3.4 ± 0.19

**Table 6 polymers-13-00740-t006:** Flexural properties of PLA/PBAT/PBS/nano talc composite 3D prints.

PLA/PBAT/PBS/nano talc	Flexural Strength (MPa)		Flexural Modulus (GPa)		Deflection Distance (mm)	
	Horizontal	Vertical	Horizontal	Vertical	Horizontal	Vertical
70/30/0/10	62.7 ± 1.29	16.8 ± 0.69	1.9 ± 0.14	1.1 ± 0.06	12.1 ± 0.19	1.3 ± 0.02
70/0/30/10	83.7 ± 1.23	24.2 ± 4.22	2.6 ± 0.01	1.5 ± 0.00	8.8 ± 1.53	1.4 ± 0.24
70/10/20/10	73.9 ± 1.07	26.0 ± 1.41	2.3 ± 0.12	1.5 ± 0.04	13.1 ± 0.09	1.6 ± 0.09

**Table 7 polymers-13-00740-t007:** Void area and properties of PLA/PBAT/PBS/nano talc composite 3D prints.

PLA/PBAT/PBS/nano talc	Void Area (mm^2^)		η^* 1^ (kPa·s)	T_c_ (°C)	1st *X_c_* (%)
	Horizontal	Vertical			
70/30/0/10	0.07	0.01	4.0	114.0	37.7
70/0/30/10	0.08	0.01	7.1	116.9	52.8
70/10/20/10	0.05	0.02	3.8	113.5	47.3

^1^ η^*^ is complex viscosity was recorded at 1 rad/s.

**Table 8 polymers-13-00740-t008:** Surface roughness average (R_a_) of PLA/PBAT/PBS/nano talc composites 3D prints.

PLA/PBAT/PBS/nano talc	Width Side	Thickness Side
	R_a_ (µm)	R_a_ (µm)
Horizontal		
70/30/0/10	16.0 ± 0.19	9.7 ± 0.09
70/0/30/10	17.3 ± 0.15	5.5 ± 0.20
70/10/20/10	15.2 ± 0.24	4.6 ± 0.27
Vertical		
70/30/0/10	17.5 ± 0.04	23.1 ± 0.34
70/0/30/10	18.4 ± 0.18	30.0 ± 0.49
70/10/20/10	17.7 ± 0.05	24.4 ± 0.78

**Table 9 polymers-13-00740-t009:** Dimensional accuracy of PLA/PBAT/PBS/nano talc composites 3D prints.

PLA/PBAT/PBS/nano talc	Width (mm)	Deviation (%)	Length (mm)	Deviation (%)	Thickness (mm)	Deviation (%)
Horizontal						
70/30/0/10	10.08 ± 0.03	−0.75	59.87 ± 0.02	0.22	1.97 ± 0.03	1.35
70/0/30/10	9.98 ± 0.07	0.20	59.49 ± 0.01	0.84	2.03 ± 0.03	−1.35
70/10/20/10	10.03 ± 0.04	−0.28	59.72 ± 0.02	0.47	2.01 ± 0.03	−0.56
Vertical						
70/30/0/10	10.03 ± 0.05	−0.32	59.92 ± 0.07	0.14	2.03 ± 0.02	−1.54
70/0/30/10	9.95 ± 0.06	0.55	59.96 ± 0.05	0.06	2.13 ± 0.04	−6.45
70/10/20/10	9.97 ± 0.06	0.34	59.93 ± 0.02	0.11	2.07 ± 0.02	−3.65

Remark: Original dimension from CAD: width = 10 mm, length = 60 mm, thickness = 2 mm.

## Data Availability

The data presented in this study are available on request from the corresponding author.

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
