# Peer review of "Improvement of Interlayer Adhesion and Heat Resistance of Biodegradable Ternary Blend Composite 3D Printing"

_polymers, 2021, doi:10.3390/polym13050740_

Round 1

Reviewer 1 Report

The presented study entitled "Improvement of Interlayer Adhesion and Heat Resistance of Biodegradable Ternary Blend Composites 3D Printing" refers to the subject of FDM printing technique. Since the properties of PLA/PBS, PLA/PBAT and ternary blends were already investigated, the subject of research is not novel. However, taking into account the use of prepared materials in the FDM method, the work acquires innovative values. A noteworthy fact is the performance of many comprehensive tests characterizing the properties of the obtained samples. The research plan itself has been developed very carefully, the research methods are well described. Presentation of the results, their description and analysis are also of high quality. In my opinion, the presented manuscript can be consider as the research article and is suitable for publication in POLYMERS journal.

Author Response

Thank you very mcuh for your kind comments and support the manuscript.

Reviewer 2 Report

I have reviewed a manuscript entitled “Improvement of Interlayer Adhesion and Heat Resistance of Biodegradable Ternary Blend Composites 3D Printing”. It is an interesting work focusing on the binary and ternary blend composite filament from PLA/PBAT/PBS/nano talc FDM 3D printing. They provide almost comprehensive results supporting their claim. I think it is suitable for publication after addressing the following comments:

  • Figure 5 shows shear-thinning behavior. I would suggest quantifying the extent of shear-thinning by using the power-law model. The following might be helpful:

https://pubs.acs.org/doi/abs/10.1021/acsabm.0c00169

https://pubs.rsc.org/fi/content/articlehtml/2020/nr/d0nr02581j

https://onlinelibrary.wiley.com/doi/full/10.1002/pat.4654

  • Could you please clarify the morphology of the ternary polymer blend? It this core-shell structure of dispersed droplets?
  • I would suggest to add a paragraph and support the observed morphology with mathematical models. The following might be helpful:

https://www.sciencedirect.com/science/article/pii/S0014305714000275

  • Please consider editing the language as sometimes it is hard to follow the text.

Author Response

Thank you very much for your kind comments. Please see the response in the attachment.

Round 2

Reviewer 2 Report

It is a well-written and interesting manuscrip. Respected authors addressed all mentioned comments in details. It is suitable for the publication in the present form.